# Impact of COVID-19 on health services utilization in mainland China and its different regions based on S-ARIMA predictions

Xiangliang Zhang[1,2☉], Rong Yin[1,2☉], Meng Zheng[1,2], Di Kong[1,2], Wen Chen[1,2]*

**1** Department of Medical Statistics, School of Public Health, Sun Yat-sen University, Guangzhou, China,
**2** Center for Migrant Health Policy, Sun Yat-sen University, Guangzhou, China

☉ These authors contributed equally to this work.
* chenw43@mail.sysu.edu.cn

**Data Availability Statement:** All data used in this research are open to the public. Data about health services utilization are publicly available from the websites of Statistical Information Center of the

## Abstract

Global health services are disrupted by the COVID-19 pandemic. We evaluated extent and duration of impacts of the pandemic on health services utilization in different economically developed regions of mainland China. Based on monthly health services utilization data in China, we used Seasonal Autoregressive Integrated Moving Average (S-ARIMA) models to predict outpatient and emergency department visits to hospitals (OEH visits) per capita without pandemic. The impacts were evaluated by three dimensions:1) absolute instant impacts were evaluated by difference between predicted and actual OEH visits per capita in February 2020 and relative instant impacts were the ratio of absolute impacts to baseline OEH visits per capita; 2) absolute and relative accumulative impacts from February 2020 to March 2021; 3) duration of impacts was estimated by time that actual OEH visits per capita returned to its predicted value. From February 2020 to March 2021, the COVID-19 pandemic reduced OEH visits by 0.4676 per capita, equivalent to 659,453,647 visits, corresponding to a decrease of 15.52% relative to the pre-pandemic average annual level in mainland China. The instant impacts in central, northeast, east and west China were 0.1279, 0.1265, 0.1215, and 0.0986 visits per capita, respectively; and corresponding relative impacts were 77.63%, 66.16%, 44.39%, and 50.57%, respectively. The accumulative impacts in northeast, east, west and central China were up to 0.5898, 0.4459, 0.3523, and 0.3324 visits per capita, respectively; and corresponding relative impacts were 23.72%, 12.53%, 13.91%, and 16.48%, respectively. The OEH visits per capita has returned back to predicted values within the first 2, 6, 9, 9 months for east, central, west and northeast China, respectively. Less economically developed areas were affected for a longer time. Safe and equitable access to health services, needs paying great attention especially for undeveloped areas.

## 1. Introduction

The coronavirus disease 2019 (COVID-19) pandemic has become a global crisis and is interfering with health systems in almost every country [1, 2]. Especially in the early phase, hospitals

National Health Commission of the People Republic of China (http://www.nhc.gov.cn/mohwsbwstjxxzx/s2906/new_list.shtml), and data about populations from the websites of National Bureau of the People Republic of China (http://www.stats.gov.cn/tjsj/pcsj/). People can get access to these data without neither registration nor application. Authors are not precluded from accessing data in the study, and they accept responsibility to submit for publication.

**Funding:** This work is supported by National Natural Science Foundation of China (72074229). This funding source had no role in the design of this study and will not have any role during its execution, analyses, interpretation of the data, or decision to submit results.

**Competing interests:** The authors declare no conflicts of interests.

were overwhelmed with the daily increasingly number of COVID-19 cases. Health systems in some countries even collapsed due to inadequate clinical staff and insufficient medical facilities, thus, routine hospital activities had to make way for treating COVID-19 patients, which greatly reduced health services for non COVID-19 patients [3].

Since 2020, disruptions in health services have been widespread across the globe [4]. For example, the number of hospital admissions across Canada declined by 36% from March to June 2020 [5]; inpatient dermatology visits in China witnessed dramatic disease-specific reductions by nearly 61% in 2020 [6]; and visits for diabetes during 2020 declined by 39% in the Democratic Republic of the Congo [7]. Even under the normalization of epidemic prevention and control, global health systems are still being challenged [8]. Although a tough task, maintaining continuity of essential health services and getting health services utilization back to normal have always been ranked on the priority list for the world in the post COVID-19 era [9].

China, with a vast territory and diverse demography, socioeconomic status, and healthcare systems, is very likely to experience heterogeneous impacts of the COVID-19 pandemic on health services utilization in different regions. However, almost all available research focused on short-term impacts of COVID-19 on visits in specific healthcare settings and in limited regions. There is also a lack of research on longer-term impacts of the pandemic on health systems that would help guide future health policy development and resource allocation and inspire how we can build a better health delivery system in the post-pandemic era.

To our knowledge, the COVID-19 impacts on health services utilization were evaluated primarily by comparing data for a certain time period in 2020 with data for the corresponding period in 2019 [6, 10, 11], ignoring the intrinsic time trend of these data. In fact, time series analysis is a better approach to dealing with time dependent variables (e.g. monthly outpatient visits). As one of the classical forecasting models in time series analyses, Seasonal Autoregressive Integrated Moving Average (S-ARIMA), has shown its good performance in COVID-19 cases predictions [12], but it has not been applied in forecasting health services utilization yet.

In addition to the short impact period of the pandemic that has been the focus of existing studies, its measurement metrics can also be optimized. In general, where there is a larger population size, there are more health sources allocations, which lead to greater health services utilization. To eliminate confounding effects brought by population sizes and make it more comparable, "outpatient visits per capita" and its relative metrics, for instance, would be better indicators in comparing health services utilization across regions, but similar indicators have not been used in previous research.

Therefore, in current study, we introduced a counterfactual thinking by using S-ARIMA predictions in different provinces and economically developed regions of mainland China and then estimated the instant and accumulative impact of COVID-19 on outpatient and emergency visits to hospitals (OEH visits) per capita based on the differences between monthly routine monitoring data and predicted data from February 2020 to March 2021 and to measure how long the impact lasted. The findings could help to find distinct impact patterns on health services utilization and try to respond to calls [13, 14] from the World Health Organization, to build health systems resilience for universal health coverage and health security during the COVID-19 pandemic and beyond.

## 2. Methods

### 2.1 Data sources

We extracted data of monthly outpatient and emergency department visits, and monthly inpatients discharged from January 2016 to March 2021, from routine monitoring statistics

published by the Statistical Information Center of the National Health Commission of China. These online data are publicly available and provide national aggregate in mainland China grouped by facility types (hospitals/township health centers) and by provinces. Besides, we collected provincial, regional and national population data of 2010 and 2020 via the sixth [15] and the seventh [16] national census reports.

## 2.2 Measurement

We collected data on four types of health services utilization per capita, namely 1) outpatient and emergency visits to hospitals (OEH visits) per capita: the sum of outpatient and emergency department visits per capita to urban healthcare facilities; 2) outpatient and emergency department visits to township health center (OET visits) per capita: the sum of outpatient and emergency department visits per capita to rural healthcare facilities; 3) inpatient discharged from hospitals (IH visits) per capita: the number of inpatient visits per capita to urban healthcare facilities; and 4) inpatient discharged from township health centers (IT visits) per capita: the number of inpatient visits per capita to rural healthcare facilities. Since OEH visits per capita always account for the largest proportion, 75.66%, of health services utilization in mainland China, and only regional differences in OEH visits per capita was observed based on our exploratory analyses (see *3.1 Overview of four types of health services utilization per capita in China* in the Results Section). We therefore merely took interest in the impacts on OEH visits per capita in our final analyses.

To measure health services utilization per capita, we first estimated populations in mainland China and in each province for the years 2011–2019 and 2021 assuming that there was a constant annual growth rate in population from 2010 to 2021. By dividing monthly visits by the corresponding populations, we obtained monthly services utilization per capita of 31 provinces. In a similar way, we measured monthly services utilization per capita of four regions of different economic status (west, east, central and northeast China) [17], as well as the average level nationwide. Specific geographical locations of 31 provinces and 4 regions were shown in S1 Fig.

## 2.3 Statistical analyses

**2.3.1 Modeling.** We conducted time series analyses to evaluate impacts of the COVID-19 pandemic on OEH visits per capita by establishing S-ARIMA models. In our analyses, the total time series went through from January 2016 to March 2021. For each province and region, as well as the country, 45 data points before the pandemic, namely from January 2016 to January 2020 (referred as the original series), were included in the trained model to forecast 13 data points after COVID-19 outbreak, namely from February 2020 to March 2021 (referred as the predicted series). We chose November 2019 as the endpoint of our model training for Hubei province and January 2020 for other provinces for: 1) Data in every December were unpublished, which were hard to fit with monthly regularity; 2) except for Hubei province, emergency responses to the COVID-19 were implemented in the late January 2020 and had limited impacts on overall OEH visits in January [18]; and 3) for Hubei province, the COVID-19 was first broke out in Wuhan city in December 2019 and several local measures had been taken in January 2020 [18–20]. To separate seasonal effects and fit the fluctuation of time series, we set a S-ARIMA model with six parameters S-ARIMA $(p, D, q) \times (p_s, D_s, q_s)$, and the sequence value $y_t$ was estimated by following functions:

$$\phi(L)\Phi(L)(1 - L)^D(1 - L^s)^{D_s}y_t = c + \theta(L)\Theta(L)\varepsilon_t,$$

with

$$\phi(L) = \left(1 - \phi_1 L - \ldots - \phi_p L^p\right), \ \Phi(L) = 1 - \Phi_{p_s} L^{p_s},$$

$$\theta(L) = \left(1 + \theta_1 L + \ldots + \theta_q L^q\right), \ \Theta(L) = \left(1 + \Theta_{q_s} L^{q_s}\right).$$

Here, the notation $L^i y_t = y_{t-i}$. The time sequence variable $t$ ranged from 1 (2016/01) to 45 (2020/01), except Hubei ranging from 1 (2016/01) to 44 (2019/11). In fitting process, we used 11 ($D_s$) as a period to differentiate seasonality. By testing different orders of ($p$, $D$, $q$, $p_s$, $D_s$, $q_s$), we chose the optimal model for each province based on following criteria: 1) minimization of sum of AIC and BIC; 2) the nearest Euclidean distance between (F-S, P-S) and (1,1) with comprehensive considerations of the trend consistency of predicted and original series (S1 Table).

MATLAB (version 9.6.0.10727) and its built-in application Econometrics Toolbox (version 5.2 R2019a) were applied in our statistical analyses. Separate training instruction files in the Supplementary materials were generated for each model training process. To visualize impacts of COVID-19 on OEH visits per capita of 31 provinces, Tableau Desktop (version 20021.3) and ArcGIS (version 10.8.1) were used to create the figures.

**2.3.2 Evaluations.** Based on S-ARIMA models, we predicted national, regional and provincial values of monthly OEH visits per capita from February 2020 to March 2021, with 95% confidence intervals correspondingly, if there were no COVID-19 pandemic. Then, by comparing the actual figures and predicted figures, we evaluated the impact of the pandemic from the following three aspects.

*2.3.2.1 Evaluations of instant impacts.* **Instant impacts-absolute**: For each province and region, as well as the country, by calculating the difference between a predicted value and its corresponding observed value of each month, we got the loss of monthly OEH visits per capita. The loss of monthly OEH visits per capita in February 2020 reflected the absolute instant impact.

**Instant impacts-relative**: The relative instant impact was defined as the ratio of the absolute instant impact to the baseline level. Regarding the definition of baseline, we chose the mean OEH visits per capita in the 1 year before the pandemic (**2019/01-2020/01**) as the baseline.

*2.3.2.2 Evaluations of accumulative impacts.* **Accumulative impacts-absolute**: For each province and region, as well as the country, adding all values of loss of monthly OEH visits per capita from February 2020 to March 2021, we obtained the total loss of OEH visits per capita, which indicated the absolute accumulative impacts of COVID-19 pandemic on health services utilization.

**Accumulative impacts-relative**: Similar to the evaluation of relative instant impact, relative impacts (relative to the baseline) for each month from February 2020 to March 2021 were calculated and then were averaged over such a period. The relative accumulative impact was denoted as the ratio of the average impact to the baseline level. Likewise, the mean monthly OEH visits per capita 1 year before the pandemic (**2019/01-2020/01**) was referred to as the baseline.

*2.3.2.3 Evaluations of recovery time.* We defined the recovery time as time it took for the number of OEH visits per capita in a region to return to its predicted value assuming no COVID-19 pandemic. To calculate the recovery time, we examined whether there was significant difference between predicted time series and observed time series at each predicted time point and thereafter by Wilcoxon matched-pairs signed ranks sum test. We identified the first

non-significant ($P$>0.05) time point and denoted it as $T_A$. We then counted months between February 2020 and $T_A$ as the recovery time.

## 3. Results

### 3.1 Overview of four types of health services utilization per capita in China

Among four types of health services utilization, we only observed wide differences in OEH visits per capita among all provinces in mainland China (Fig 1a and 1b). Before February 2020, of 31 provincial administrative units, Beijing had the highest monthly OEH visits per capita (above 0.4000), while Tibet had the lowest (less than 0.0500). The COVID-19 breakout not only brought a sudden decline in OEH visits per capita for all provinces, but narrowed the gap of OEH visits per capita across the nation in February 2020. With time went by, the huge variation reappeared.

### 3.2 Instant impacts of COVID-19

In general, the absolute instant impact of COVID-19 in central China (0.1279 visits per capita) and northeast China (1.1265 visits per capita) was obviously greater than that in other regions (East: 1215 visits per capita; West: 0.0985 visits per capita) (Table 1). Sorted by the absolute instant impacts of COVID-19, the top three provincial administrative units were Beijing, Shanghai and Zhejiang (all locate in east China), with the loss of monthly OEH visits per capita of 0.3306, 0.2804 and 0.2105 in February 2020, respectively, while Qinghai (locates in west China), suffered the smallest loss of monthly OEH visits per capita (0.0465 visits per capita). However, it was the central region, instead of the east, that suffered the largest instant impact on the OEH visits per capita in February 2020. To our surprise, Hubei Province (Fig 2a), with Wuhan as the epicenter of the COVID-19 in China, had the monthly OEH visits loss per capita of 0.1586, ranked the fifth in February 2020. The 95% confidence intervals of absolute instant impacts in each province and region per month were presented in Table 1.

In terms of the relative instant impacts, Heilongjiang (locates in Northeast China), became the most heavily hit province in the first month of pandemic, which is closely followed by Hubei (locates in central China), with a cliff-like drop of 78.78% and 78.50%, respectively (Fig 2b). Besides, their adjacent provinces, like Shaanxi and Henan, Jilin and Nei Mongol, also witnessed a considerable loss of OEH visits per capita (above 60%). As a whole, in the early stage, the percentage change of monthly OEH visits per capita was up to 57.59% for mainland China, and it fell by 77.63%, 66.16%, 50.57%, and 44.39% companied with the pre-pandemic level for central, northeast, west and, east China, respectively (Table 1). The 95% confidence intervals of relative instant impacts in each province and region per month were presented in Table 1.

### 3.3 Accumulative impacts of COVID-19

Generally, the national total loss of the monthly OEH visits per capita was up to 0.4676 visits per capita, equivalent to a reduction of 659,453,647 OEH visits. The absolute accumulative impact of COVID-19 on OEH visits per capita was concentrated in the northeastern (0.5859 visits per capita) and eastern provinces (0.4459 visits per capita) (Fig 3a). In detail (Table 1), Beijing, as the most affected provincial administrative unit, had a total loss of 2.5187 OEH visits per capita, 5.39 times of the national average level (0.4676 visits per capita), which was followed by Shanghai, with a total loss of 1.3501 OEH visits per capita, 2.88 folds of the national average level. By contrast, Anhui, located in Central China, experienced the least total loss of OEH visits per capita (0.0173 visits per capita).

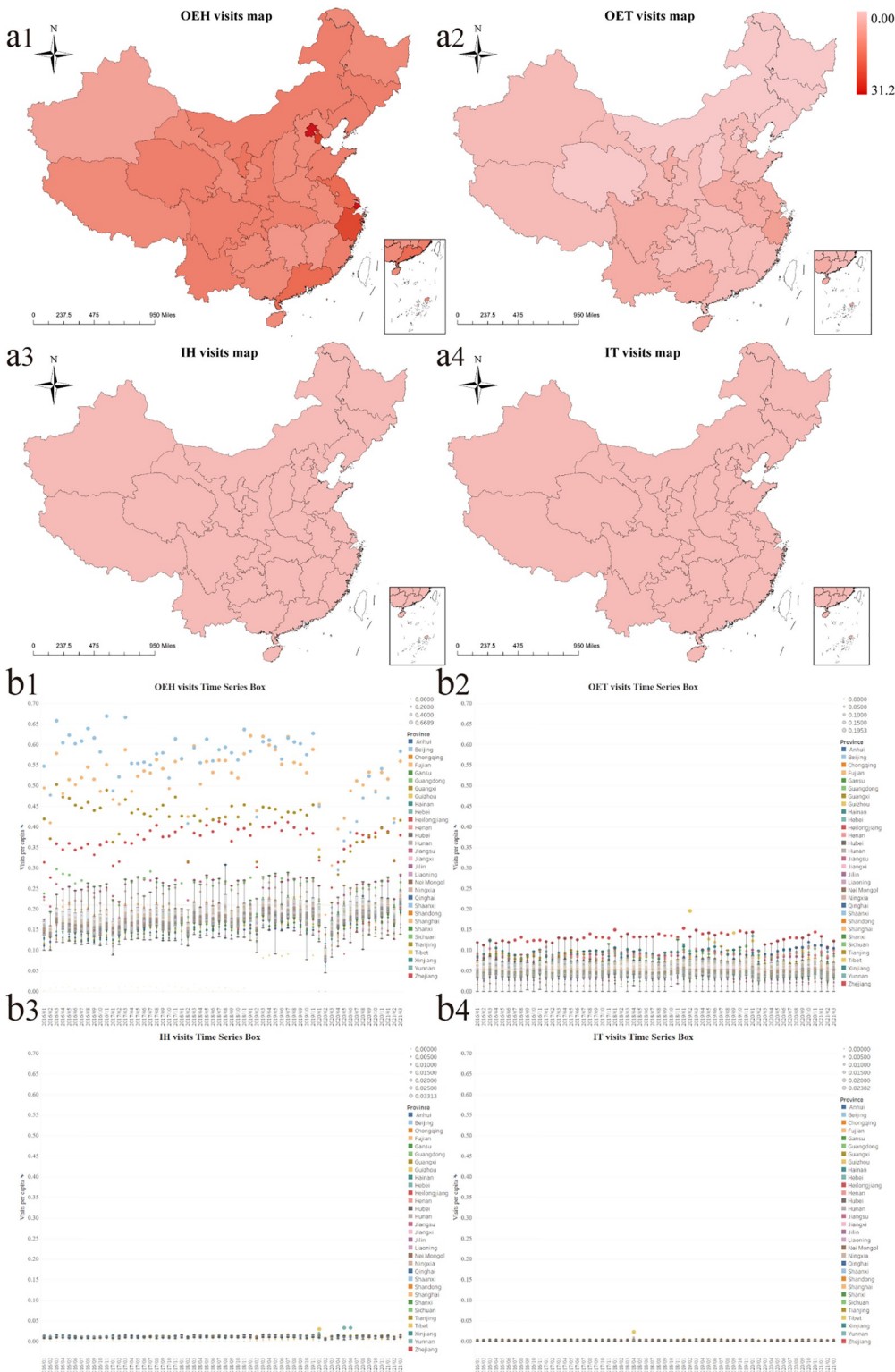

**Fig 1. Four types of health services utilization per capita with their regional difference in mainland China.** (a1, a2, a3, a4) These panels show the sum of visits per capita in four healthcare settings in mainland China from January 2016 to March 2021. They represent the number of OEH visits per capita, OET visits per capita, IH visits per capita, and IT visits per capita, respectively. Each color block reflects the visits per capita of each province. (b1, b2, b3, b4) These panels show actual monthly visits per capita in four healthcare settings of 31 provinces in mainland China from

January 2016 to March 2021. They represent actual monthly visits per capita of OEH visits per capita, OET visits per capita, IH visits per capita, and IT visits per capita, respectively. Different colors represent different provinces. The size of dots indicates the monthly visits per capita. Maps were created using ArcGIS by ESRI version 10.8 (http://www.esri. com) [21]. Base map sources were at http://bzdt.ch.mnr.gov.cn/ and https://dataverse.harvard.edu/dataset.xhtml? persistentId=doi:10.7910/DVN/DBJ3BX. Detailed data utilized in the creation of this figure were from the Statistical Information Center of the National Health Commission of China (http://www.nhc.gov.cn/mohwsbwstjxxzx/s2906/ new_list.shtml).

**Table 1. Impacts of OEH visits per capita due to COVID-19.**

| Region/Province | Instant impacts | | Accumulative impacts | |
|---|---|---|---|---|
| | Absolute | Relative (%) | Absolute | Relative (%) |
| China | 0.1256 (0.1187,0.1325) | 57.59 (54.42,60.75) | 0.4676 | 16.48 |
| Central | 0.1279 (0.1220,0.1337) | 77.63 (74.08,81.18) | 0.3324 | 15.52 |
| Anhui | 0.0743 (0.0652,0.0834) | 47.94 (42.08,53.80) | 0.0173 | 0.86 |
| Henan | 0.1167 (0.1116,0.1218) | 62.66 (59.93,65.39) | 0.5088 | 21.01 |
| Hubei | 0.1585 (0.1502,0.1669) | 78.50 (74.35,82.64) | 0.5632 | 19.91 |
| Hunan | 0.0729 (0.0683,0.0775) | 51.23 (48.00,54.46) | 0.1440 | 7.79 |
| Jiangxi | 0.0748 (0.0702,0.0794) | 50.54 (47.44,53.64) | 0.2615 | 13.58 |
| Shanxi | 0.0503 (0.0441,0.0566) | 33.62 (29.44,37.80) | -0.1674 | -8.59 |
| East | 0.1215 (0.1126,0.1303) | 44.39 (41.16,47.61) | 0.4459 | 12.53 |
| Beijing | 0.3306 (0.3019,0.3593) | 57.83 (52.80,62.85) | 2.5187 | 33.89 |
| Fujian | 0.0630 (0.0552,0.0707) | 31.23 (27.39,35.06) | 0.1394 | 5.31 |
| Guangdong | 0.1052 (0.0928,0.1175) | 39.86 (35.18,44.54) | 0.5939 | 17.31 |
| Hainan | 0.1005 (0.0912,0.1098) | 57.68 (52.34,63.02) | 0.2292 | 10.12 |
| Hebei | 0.0729 (0.0673,0.0785) | 43.23 (39.92,46.54) | 0.0820 | 3.74 |
| Jiangsu | 0.1273 (0.1148,0.1398) | 50.79 (45.81,55.77) | 0.3684 | 11.30 |
| Shandong | 0.0863 (0.0790,0.0936) | 43.06 (39.41,46.71) | 0.1981 | 7.60 |
| Shanghai | 0.2803 (0.2472,0.3135) | 50.06 (44.15,55.98) | 1.3501 | 18.54 |
| Tianjin | 0.2049 (0.1824,0.2273) | 49.15 (43.77,54.53) | 1.2867 | 23.74 |
| Zhejiang | 0.2105 (0.1959,0.2250) | 55.29 (51.47,59.10) | 0.4585 | 9.26 |
| Northeast | 0.1265 (0.1200,0.1330) | 66.16 (62.78,69.55) | 0.5898 | 23.72 |
| Heilongjiang | 0.1345 (0.1297,0.1394) | 78.78 (75.94,81.62) | 0.6217 | 27.99 |
| Jilin | 0.1157 (0.1081,0.1233) | 60.07 (56.14,64.01) | 0.6400 | 25.56 |
| Liaoning | 0.0973 (0.0899,0.1048) | 47.27 (43.64,50.89) | -0.4188 | -15.64 |
| West | 0.0985 (0.0932,0.1038) | 50.57 (47.85,53.28) | 0.3523 | 13.91 |
| Chongqing | 0.1007 (0.0943,0.1072) | 49.50 (46.34,52.65) | 0.2767 | 10.45 |
| Gansu | 0.0839 (0.0769,0.0909) | 50.46 (46.23,54.68) | 0.4372 | 20.21 |
| Guangxi | 0.0809 (0.0747,0.0871) | 44.36 (40.95,47.77) | 0.3213 | 13.55 |
| Guizhou | 0.0566 (0.0525,0.0608) | 34.44 (31.90,36.98) | 0.1444 | 6.75 |
| Nei Mongol | 0.1164 (0.1073,0.1255) | 61.69 (56.86,66.53) | 0.4067 | 16.58 |
| Ningxia | 0.1014 (0.0916,0.1112) | 40.52 (36.61,44.44) | 0.3345 | 10.28 |
| Qinghai | 0.0465 (0.0270,0.0659) | 24.45 (14.22,34.68) | 0.0707 | 2.86 |
| Shaanxi | 0.1387 (0.1312,0.1462) | 67.13 (63.51,70.76) | 0.4754 | 17.69 |
| Sichuan | 0.1077 (0.1009,0.1146) | 49.76 (46.6,52.92) | 0.4704 | 16.71 |
| Tibet | - | - | - | - |
| Xinjiang | 0.0760 (0.0680,0.0840) | 44.80 (40.11,49.50) | -0.0162 | -0.74 |
| Yunnan | 0.0743 (0.0694,0.0793) | 35.88 (33.50,38.27) | 0.2181 | 8.10 |

The 95% confidence intervals of instant impacts in each province and region were presented in the brackets of the 2nd and 3rd column and there were no interval estimations for accumulative impacts.

-: No prediction and no data.

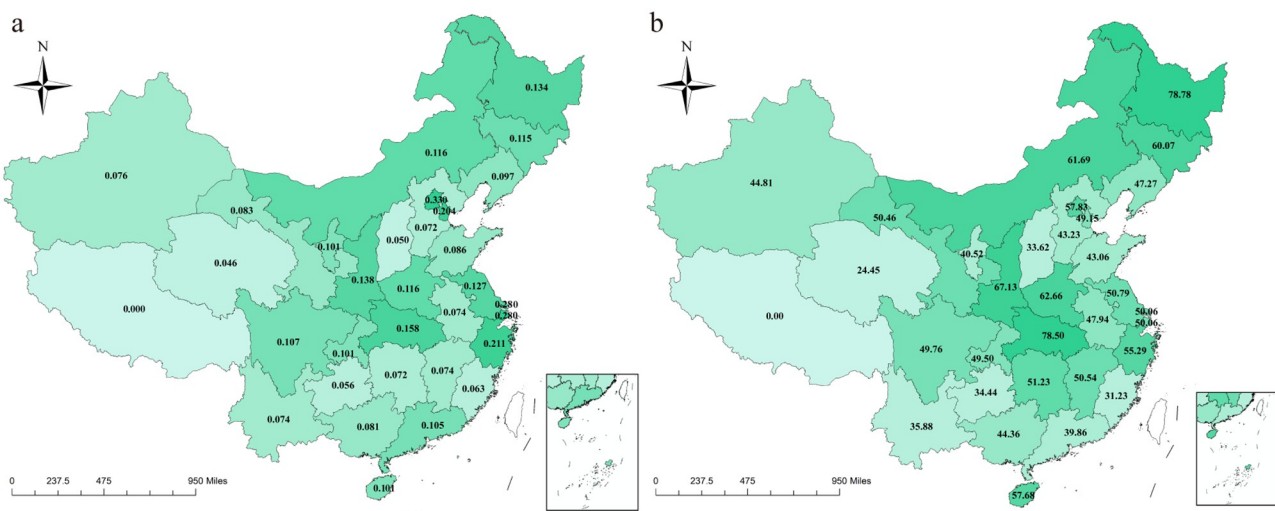

**Fig 2. Instant impacts of COVID-19 on OEH visits per capita in mainland China.** (a) This panel depicts the absolute instant impact of COVID-19 on monthly OEH visits per capita in February 2020 by province. Different background colors represent different regions of mainland China. The darkness of each block indicates the loss of monthly OEH visits per capita in February 2020 of each province (Tibet was excluded because of missing data). (b) This panel depicts the relative instant impact in February 2020 by province. The darkness of each block indicates the relative reduction on monthly OEH visits per capita in February 2020 of each province, compared to the pre-pandemic level. Tibet was excluded because of missing data. Maps were created using ArcGIS by ESRI version 10.8 (http://www.esri.com) [21]. Base map sources were at http://bzdt.ch.mnr.gov.cn/ and https://dataverse.harvard.edu/dataset.xhtml?persistentId=doi:10.7910/DVN/DBJ3BX. Detailed data utilized in the creation of this figure were from the Statistical Information Center of the National Health Commission of China (http://www.nhc.gov.cn/mohwsbwstjxxzx/s2906/new_list.shtml).

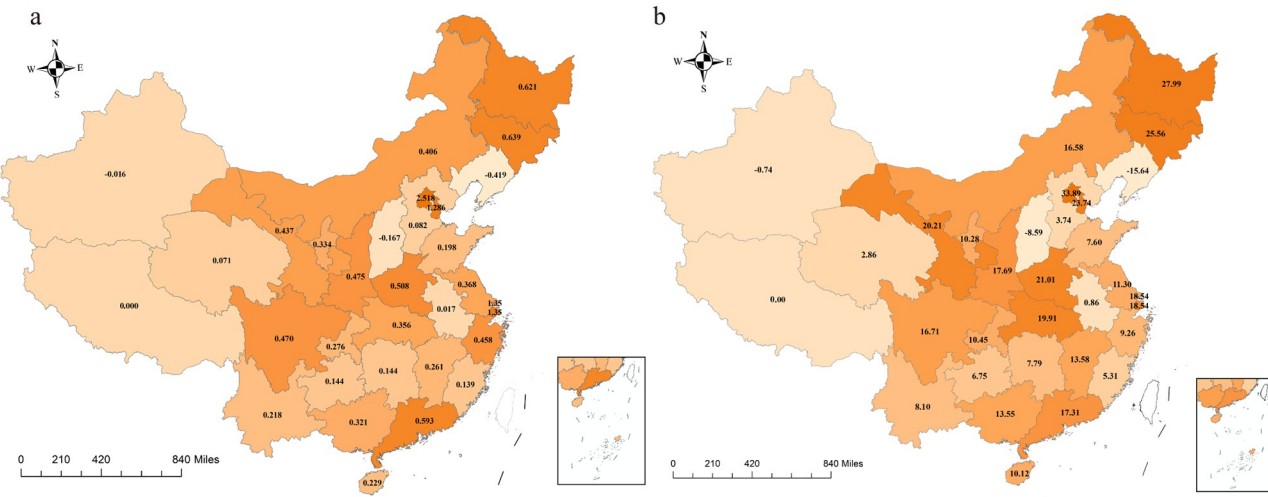

**Fig 3. Accumulative impacts of COVID-19 on OEH visits per capita in mainland China.** (a) This panel plots the total loss of OEH visits per capita from February 2020 to March 2021 by province. The darkness of each block represents the total loss of total OEH visits per capita of each province from February 2020 to March 2021 Tibet was excluded because of missing data. (b) This panel plots the relative loss of OEH visits per capita from February 2020 to March 2021 by province. The darkness of each block indicates relative loss of OEH visits per capita from February 2020 to March 2021 of each province, compared to the pre-pandemic level. Tibet was excluded because of missing data. Maps were created using ArcGIS by ESRI version 10.8 (http://www.esri.com) [21]. Base map sources were at http://bzdt.ch.mnr.gov.cn/ and https://dataverse.harvard.edu/dataset.xhtml?persistentId=doi:10.7910/DVN/DBJ3BX. Detailed data utilized in the creation of this figure were from the Statistical Information Center of the National Health Commission of China (http://www.nhc.gov.cn/mohwsbwstjxxzx/s2906/new_list.shtml).

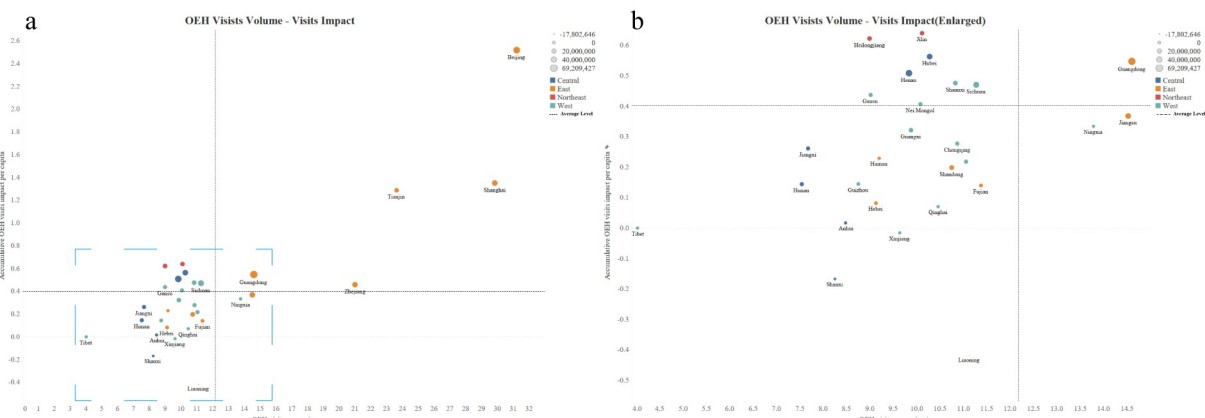

**Fig 4. Pre-pandemic OEH visits per capita and its corresponding accumulative impacts of COVID-19 for different provinces.** (a) The scatter diagram shows the association between the volume of OEH visits per capita before the pandemic and absolute accumulative impacts on OEH visits per capita for each province. The horizontal axis measures the sum of OEH visits per capita from January 2016 to November 2019, while the vertical axis measures the total loss of OEH visits per capita from February 2020 to March 2021. The horizontal and vertical gray dotted lines respectively show the average loss of visits per capita and average visits per capita nationwide. One dot represents one province. The same color means belonging to the same region. The size of each dot represents the total loss of OEH visits per capita. (b) This panel is an enlarged view of the part labeled by the blue dashed square in Panel a).

Relative to the average annual level before the pandemic, the average monthly OEH visits per capita from February 2020 to March 2021, decreased by 15.52% in mainland China. Similarly, Beijing (locates in east China), was the top administrative provincial unit with the most relative loss (33.89%). Northeast China was also the region that suffered the greatest percentage change in OEH visits per capita. For provinces in northeast China, e. g. Heilongjiang and Jilin, a large decrease of 27.99% and 25.56%, respectively, on OEH visits per capita was observed throughout our research period.

As shown in Fig 4a, for the majority of the provinces in mainland China, the larger number of OEH visits per capita before the pandemic, the more total loss of per capita OEH visits per capita during the COVID-19 pandemic. East China (the 1st quadrant of Fig 4a), the previously most visited region, experienced the greatest loss of OEH visits per capita after the COVID-19 outbreak. Notably, as shown in the 2nd quadrant of Fig 4b, most northeastern provinces (Jilin and Heilongjiang), some western provinces (Gansu, Shaanxi and Sichuan), and several central provinces (Hubei and Henan) had low levels of OEH visits per capita prior to the pandemic, but experienced high levels of decline in OEH visits per capita during the pandemic.

### 3.4 Recovery time

Fig 5 showed all the regions were struck heavily by the COVID-19 in February 2020. The strike diminished with time, but the capability of recovering from the pandemic varied by regions. As listed in Table 2, it took almost nine months for both the western and northeastern region to eliminate the impact of COVID-19, while it took about six months for the central region. The eastern region was the fastest to get rid of the thrust of the pandemic since it had a non-negligible difference of OEH visits per capita for just two months.

## 4. Discussion

Due to insufficient emergency preparedness for a pandemic of global health systems, the COVID-19 heavily hits health services utilization worldwide [8, 14]. Many studies have shown

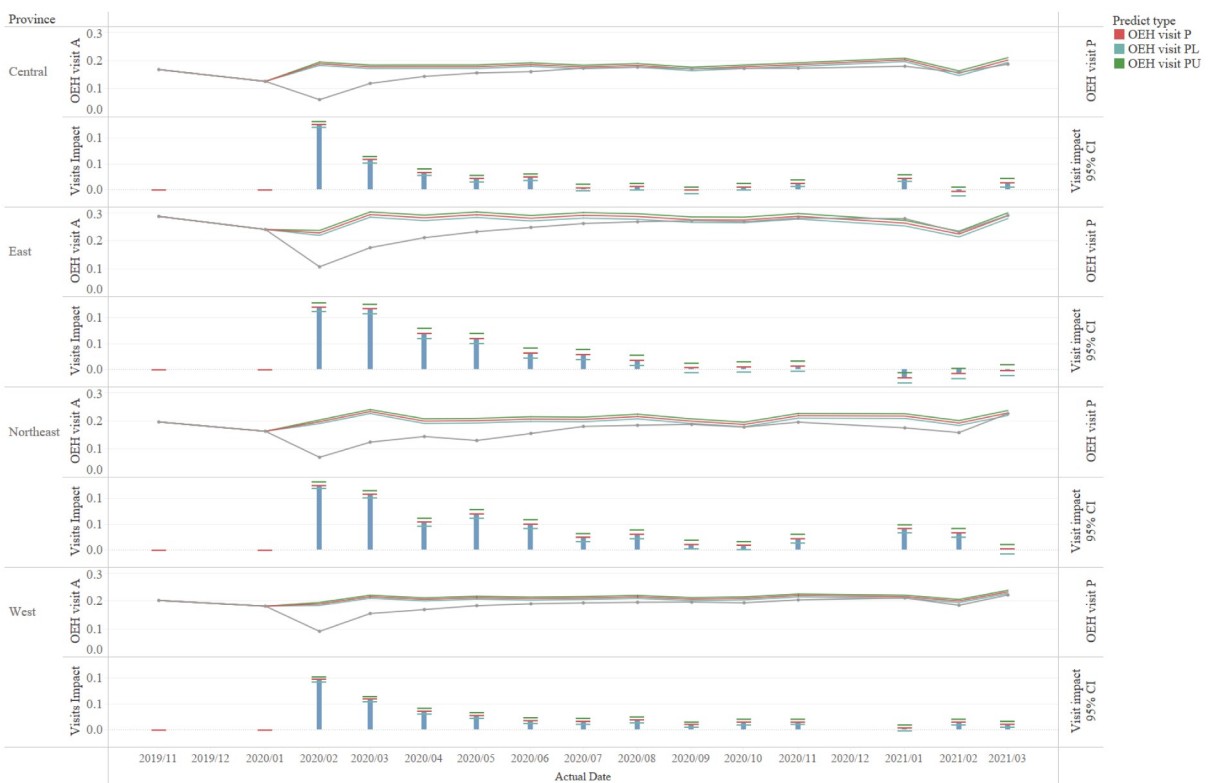

**Fig 5. Trend of impacts on the monthly OEH visits per capita by regions from October 2019 to March 2021, based on the prediction of S-ARIMA model.** The gray solid line (dot) indicates the actual number of monthly OEH visits per capita, while the solid red line (dot) represents the predicted number with its 95% upper and lower confidence limit in green and blue line, respectively. The height of each histogram and red horizontal lines (OEH visit P) shows the value of the predicted number minus corresponding actual number, with its 95% upper and lower confidence limit in green (OEH visit PU) and blue (OEH visit PL) horizontal lines, respectively.

**Table 2. Recovery time and *P* values of the Wilcoxon matched-pairs signed ranks sum test on actual and predicted time series of hospital visits per capita in mainland China.**

| Time Series Period | NRM[a] | Central | East | Northeast | West | China |
|---|---|---|---|---|---|---|
| 2020/02-2021/03 | 1 | 0.0030 | 0.0192 | 0.0015 | 0.0015 | 0.0015 |
| 2020/03-2021/03 | 2 | 0.0047 | 0.0342 | 0.0022 | 0.0022 | 0.0022 |
| 2020/04-2021/03 | 3 | 0.0076 | 0.0619 | 0.0033 | 0.0033 | 0.0033 |
| 2020/05-2021/03 | 4 | 0.0125 | 0.1141 | 0.0051 | 0.0051 | 0.0051 |
| 2020/06-2021/03 | 5 | 0.0209 | 0.2135 | 0.0077 | 0.0077 | 0.0077 |
| 2020/07-2021/03 | 6 | 0.0357 | 0.4008 | 0.0117 | 0.0117 | 0.0117 |
| 2020/08-2021/03 | 7 | 0.0630 | 0.7353 | 0.0180 | 0.0180 | 0.0180 |
| 2020/09-2021/03 | 8 | 0.1159 | 0.7532 | 0.0277 | 0.0277 | 0.0277 |
| 2020/10-2021/03 | 9 | 0.0796 | 0.5002 | 0.0431 | 0.0431 | 0.0431 |
| 2020/11-2021/03 | 10 | 0.1441 | 0.2733 | 0.0679 | 0.0679 | 0.0679 |
| 2021/01-2021/03 | 12 | 0.2850 | 0.1088 | 0.1088 | 0.1088 | 0.1088 |
| 2021/02-2021/03 | 13 | 0.6547 | 0.1797 | 0.1797 | 0.1797 | 0.1797 |

NRM: Number of recovery months.

[a]: Number of recovery months was calculated as the number of months between the start month of the time series period for the same row and January 2020.

significant decrease in medical visits for diseases other than COVID-19 in 2019–2020, further exacerbating health inequity globally [22–26]. In this study, based on counterfactual thinking, we used the differences between routine monitoring data and predicted data by S-ARIMA models to evaluate impacts of COVID-19 on four types of health services utilization per capita in mainland China from February 2020 to March 2021, and found that the pandemic only significantly cut down one of them, namely OEH visits, by 0.4676 per capita in mainland China, equivalent to a reduction of 659,453,647 visits, corresponding to a decrease of 15.52% relative to the pre-pandemic average annual level. The impacts on outpatient and emergency department services of the pandemic varied across regions with different levels of economic development. Overall, the instant impact was greatest in central China, while the largest accumulative impact was in the northeast. In addition, for regions with lower level of economic development, e.g., west China, both instant and accumulative impacts were small but its recovery time was long. As the most economically developed region in China, typical east provinces, e.g., Beijing and Shanghai, suffered greatly in the early stage of the pandemic, but its health system recovered much more quickly than any other regions.

In the last decade, China's urbanization rate has raised from 46.6% to 60.6% [27, 28], with a continuous increase in the number of hospitals and a consequent decrease in the amount of township health centers. Else, deficient health workforce and unadvanced medical equipment in township health centers made people prefer hospitals when seeking for health services. Therefore, hospitals serve as the major providers of health services in mainland China. Different from hospitals, township health centers provide routine chronic medication and ailment treatment to a relatively stable number of rural residents. So, the volume of health service utilization in township health centers (i.e., OET and IT) is insusceptible to the pandemic. The possible reason why inpatient utilization rates (i.e., IH and IT) were less flexible during the pandemic could be that patients who are advised to be hospitalized, always suffer more severe diseases or require more urgent clinical care than normal outpatients. They are less likely to delay or cancel hospitalization due to difficulties correlated with COVID-19.

As expected, the monthly OEH visits per capita of all regions plummeted down after COVID-19 broke out. In February 2020, among the four regions, central China performed worst in both the absolute and relative loss of monthly OEH visits per capita, which was reasonably explained by the pandemic severity in the very early stage. Hubei, where COVID-19 cases first emerged in mainland China, along with its neighboring provinces, e.g. Henan, Hunan, Anhui and Jiangxi (all locate in central China), held 71,122 confirmed cases which approximately accounting for 87% of national confirmed cases as of February 2020 [29–34]. In these areas, massive medical staffs were urgently reallocated to treat COVID-19 patients, only few left responsible for routine outpatient and inpatient services. Else, suspended transportation services prohibited both inter-city and intra-city flows, people thus had hardly access to sorts of health facilities.

For a long-term perspective, there was the largest total loss of OEH visits per capita in the northeast, as well as the greatest relative loss, which was likely due to the aging society [35]. In the northeast China, more than 15% of the population is over 65 years old [36]. It is the region with the most aging population. Older adults, with descending health conditions, are supposed to have more frequent health services utilization than the youth. Therefore, affected by COVID-19, reductions on OEH visits could be larger in areas with larger proportion of aged population. Furthermore, the population's reluctance to attend health facilities for fear of SARS-CoV-2 infection put the elderly at increased risk of other diseases [37–39], and their suppressed demand for non COVID-19 related care could further burden health systems and to exacerbate health inequity to a large extent [40]. Therefore, how to tackle the 'double burden' of aging and COVID-19 has developed into an urgent discussion in the post pandemic

era [41], and reducing health dependency on visits to offline hospitals via strengthened family care, tighter connections with family doctors, and developing telemedicine may help to address it.

We notice that, for east China, the impact we estimated would shrink if we considered its baseline OEH visits, which was most likely explained by the severe imbalance in health services utilization across regions before the COVID-19 broke out. In China, regional inequalities in high-quality medical resources led to remarkable healthcare bypass behaviors among residents in rural and economically underdeveloped areas [42, 43]. Correspondingly, hospitals in the east China have treated many patients from other regions. It was reported that annual OEH visits in east China had increased to 4.4 billion by the end of 2019, covering over half of that nationwide [44]. Furthermore, prosperity of economic development in eastern provinces attracted abundant migrant workers to find a job there, which, to some extent, added to its health services utilization in the baseline [45].

Meanwhile, with larger proportion of migrants in the eastern region than in others, impacts of COVID-19 on the absolute loss of OEH visits per capita were magnified in eastern provinces. First, the COVID-19 outbreak overlapped with Chinese New Year, which implied massive internal migration out of the east in early 2020, thus the number of residents in the eastern region decreased significantly during this period. Second, siphoning effect of hospitals in the eastern region was interrupted. In the context of COVID-19, strict travel restrictions for long-distance migration and complex procedures for medical visits in tertiary hospitals prompt cross-province doctor consultations. Additionally, enhanced public health interventions, such as social distancing, wearing masks and more frequent hand-washing, reduced the incidence of communicable diseases [46, 47]. For example, cases of influenza and hand-foot-and-mouth disease declined by 95.1% and 76.2% respectively in Guangdong province of China during the emergency response period [48]. Therefore, it is reasonable to believe that the decline in morbidity of communicable diseases has further reduced the local demand for health services [49].

In the previous study, Xiao suggested that the quantity of outpatient and emergency visits of nationwide China had not returned to its pre-outbreak level as of June 2020 [50]. In current study, we revealed the overall outpatient and emergency visits had recovered by October 2020. Notably, with less instant and accumulative impacts on OEH visits per capita, three-fold or more recovery time in the western region than that in the eastern region was observed during the pandemic. Inadequate emergency preparedness and clumsy response strategy, may partly explain the prolonged recovery time of health systems in such less developed regions.

As west China covers a vast area with sparse and less-urbanized populations, inferior basic infrastructures are not supportive enough for the west to swiftly respond to public emergencies. For example, with limited capacity of epidemiological investigations in west China, once COVID-19 confirmed cases appeared, local government tends to adopt tighter infection control measures to mitigate SARS-CoV-2 transmission, such as more frequent and larger scales of nucleic acid testing and longer mobility restrictions, which indicates postpones or even cancellation of planned health care to a larger extent and for a longer period [51]. Oppositely, the eastern region is more capable in such cases by more precise contact tracing and narrower blockade areas, ensuring routine health-seeking behaviors, thus showing greater resilience confronted with COVID-19. The discrepancy in capacity of emergency preparedness and response contributes to regional inequality in health services provision, and subsequently exacerbates health inequity brought by socioeconomic disparities [52]. For these reasons, promoted flexibility in emergency response is needed by means of massive vaccination, new technology and strong epidemiology teams, which helps to build more resilient health systems, as well as to eliminate health inequity especially in less developed areas [53].

Meanwhile, there are limitations in our study. First, due to missing data of Tibet, we could not set the corresponding model to evaluate impacts of the COVID-19 in this province. Second, we attempted to reasonably explain the impact heterogeneity across regions and provinces in mainland China by factors such as pandemic severity, aging population, migration and emergency response capacity, which, however, could not be proved by S-ARIMA models because these factors are difficult to measure quantitatively or can only be measured as approximately constant series of covariates. Therefore, causal relations between impact on OEH visits and such factors need further exploration. Third, although we have tested all possible S-ARIMA models for Liaoning, Shanxi and Xinjiang provinces, the total loss of OEH visits per capita based on the best model were still negative, conflicted with our commonsense, indicating that the S-ARIMA model may be not suitable for the three provinces and other models may be needed to better reflect the impacts in these provinces. However, analyses on the national and regional level were not influenced as the model fitted the merged data well. The fourth limitation is that, we got no access to the increase number of telemedicine resulting from the COVID-19. Thus, we equated the reduction of health services utilization with the loss of offline OEH visits during the pandemic, which would overestimate the real impact of COVID-19 on OEH visits.

## 5. Conclusion

From February 2020 to March 2021, the COVID-19 pandemic cut down outpatient and emergency visits to hospitals by 0.4676 per capita, equivalent to a reduction of 659,453,647 visits in mainland China, corresponding to a decrease of 15.52% relative to pre-pandemic average annual level. Areas adjacent to the epicenter of COVID-19 in February 2020, i.e., central China, suffered the greatest instant impacts, while the most aging areas, i.e., northeast China, confronted with the greatest accumulative impacts. Less economically developed areas, e.g. west China, were affected for a far longer recovery time than east China. In the post COVID-19 era, adaptations, like promoting the substitutability of offline hospitals and differentiated epidemic control strategies, should be oriented to strengthen the resilience of health systems, which ensures safe and equitable access to health services, especially for those undeveloped areas.

## Supporting information

**S1 Fig. Map of administrative division in mainland China.** The map demonstrates geographical locations of 31 provincial administrative units in mainland China. Different background colors represent different regions. Maps were created using ArcGIS by ESRI version 10.8 (http://www.esri.com) [21]. Base map sources were at http://bzdt.ch.mnr.gov.cn/ and https://dataverse.harvard.edu/dataset.xhtml?persistentId=doi:10.7910/DVN/DBJ3BX. External data utilized in the creation of this figure from the Statistical Information Center of the National Health Commission of China (http://www.nhc.gov.cn/mohwsbwstjxxzx/s2906/new_list.shtml).
(TIF)

**S1 Table. The orders chosen for each S-ARIMA model and the corresponding goodness of fit indices.** No prediction and no data. AR, D, MA, SAR, S and SMA are respectively represent the S-ARIMA model specific orders of $p$, $D$, $q$, $p_s$, $D_s$ and $q_s$. AIC and BIC respectively are Akaike Information Criterion and Bayesian Information Criterion of this model. F-S: ratio of the slope of the least binomial regression of the original and predicted series to the slope of the least binomial regression of the original series. P-S: ratio of the slope of the least binomial

regression of the predicted series to the slope of the least binomial regression of the original series.
(DOCX)

## Acknowledgments

We would like to acknowledge all members dedicated to the COVID-19 epidemic control and prevention, as well as the National Health Commission of the People Republic of China for publishing data of health services utilization open to the public.

## Author Contributions

**Conceptualization:** Xiangliang Zhang, Wen Chen.

**Data curation:** Xiangliang Zhang, Wen Chen.

**Formal analysis:** Xiangliang Zhang, Wen Chen.

**Funding acquisition:** Wen Chen.

**Investigation:** Xiangliang Zhang, Rong Yin, Wen Chen.

**Methodology:** Xiangliang Zhang, Wen Chen.

**Project administration:** Rong Yin, Meng Zheng, Di Kong, Wen Chen.

**Resources:** Wen Chen.

**Software:** Xiangliang Zhang, Wen Chen.

**Supervision:** Wen Chen.

**Validation:** Xiangliang Zhang, Wen Chen.

**Visualization:** Xiangliang Zhang, Wen Chen.

**Writing – original draft:** Xiangliang Zhang, Rong Yin, Wen Chen.

**Writing – review & editing:** Rong Yin, Meng Zheng, Di Kong, Wen Chen.

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
