## [Decision Letter · Decision Letter 0]

12 Jul 2022

PGPH-D-22-00696

Impact of COVID-19 on health services utilization in mainland China and its different regions based on S-ARIMA predictions

Dear Dr. Chen,

Thank you for submitting your manuscript to PLOS Global Public Health. After careful consideration, we feel that it has merit but does not fully meet PLOS Global Public Health’s publication criteria as it currently stands. Therefore, we invite you to submit a revised version of the manuscript that addresses the points raised during the review process.

We would like to encourage you to revise the manuscript with attention to making the main contribution of the paper more clear.

- As R1 notes, the introduction is framed as identifying where hospital systems are less resilient. Still, the discussion and the conclusion are much more about the effect of age demographics, etc. In either case, if the study aims to test these hypotheses, these should be formally incorporated and addressed in the paper’s research design section. If this is more clearly stated, this will address the concerns raised by R1 regarding the research design (ARIMA models to create a counterfactual trend and using that counterfactual to measure the impact of COVID-19) and whether this modeling strategy allows you to infer different counterfactual gaps from policies and demographic factors, or if an alternative modeling strategy would be preferrable in which factors, such as the age structure of the province, were directly incorporated into the model as exogenous (non-varying) factors. R1 also encourages you to address concerns regarding model over-fitting.

- R2 raises excellent suggestions regarding clarifications that are needed in the manuscript in several sections.

We look forward to receiving your revised manuscript.

Kind regards,

Lorena G Barberia

Section Editor

Journal Requirements:

Reviewers' comments:

Reviewer's Responses to Questions

**Comments to the Author**

1. Does this manuscript meet PLOS Global Public Health’s publication criteria? Is the manuscript technically sound, and do the data support the conclusions? The manuscript must describe methodologically and ethically rigorous research with conclusions that are appropriately drawn based on the data presented.

Reviewer #1: Yes

Reviewer #2: Partly

2. Has the statistical analysis been performed appropriately and rigorously?

Reviewer #1: Yes

Reviewer #2: No

3. Have the authors made all data underlying the findings in their manuscript fully available (please refer to the Data Availability Statement at the start of the manuscript PDF file)?

Reviewer #1: Yes

Reviewer #2: Yes

4. Is the manuscript presented in an intelligible fashion and written in standard English?

Reviewer #1: Yes

Reviewer #2: Yes

5. Review Comments to the Author

Reviewer #1: 1. On page 6, the authors say:

"114 Since OEH visits always accounts for the largest proportion of health services utilization in

115 mainland China, and only regional variations in OEH visits per capita was observed based on our

116 exploratory analyses (see section 3.1 in the Results), we merely took interest in the impacts on OEH

117 visits per capita in our later analyses."

This explanation makes it unclear why the authors decided to focus only on OEH visits. I encourage them to explain this point further.

a. As the authors say, OEH visits account for the most significant proportion of utilization. To corroborate this point, please present data on the percentage of health services utilization for each of the four measures: OEH, OET, IH, and IT.

b. I did not find section 3.1 in the Results section. Please, make sure it is in the actual manuscript.

c. From what I understood from this paragraph, regional variations were only observed in OEH visits. What could explain that such regional differences were not found in OET, IH, and IT? Please, discuss this issue in the "Discussion" section.

d. It should be "account" instead of "accounts" in the sentence "Since OEH visits always accounts (...)".

2. On page 16, the authors say:

"344 In addition, for Liaoning, Shanxi and Xinjiang, the accumulative impacts were negative,

345 conflicted with our commonsense, indicating that the S-ARIMA model may be not suitable for the

346 three provinces."

Given that the estimated S-ARIMA model did not work well for these three provinces, did the authors try other S-ARIMA specifications for those three provinces? Please, present the results for the alternative specifications for these cases.

3. When the authors present and discuss the instant and cumulative impacts of the pandemic on health care visits, I encourage them to present the estimated impacts with the respective 95% confidence interval. Since the authors are estimating the impacts, it is crucial to show the uncertainty related to the estimates, which is missing in the manuscript.

4. It is hard to read the figures. Please, improve the quality of all five figures.

Reviewer #2: Please see the attached file. Please see the attached file. Please see the attached file. Please see the attached file. Please see the attached file. Please see the attached file. Please see the attached file.

6. PLOS authors have the option to publish the peer review history of their article (what does this mean?). If published, this will include your full peer review and any attached files.

**Do you want your identity to be public for this peer review?** For information about this choice, including consent withdrawal, please see our Privacy Policy.

Reviewer #1: No

Reviewer #2: **Yes: **Robert Kubinec

---

## [Decision Letter · Decision Letter 1]

2 Dec 2022

Impact of COVID-19 on health services utilization in mainland China and its different regions based on S-ARIMA predictions

PGPH-D-22-00696R1

Dear Dr. Chen,

We are pleased to inform you that your manuscript 'Impact of COVID-19 on health services utilization in mainland China and its different regions based on S-ARIMA predictions' has been provisionally accepted for publication in PLOS Global Public Health.

Best regards,

Julia Robinson

Executive Editor

Reviewer Comments (if any, and for reference):

Reviewer's Responses to Questions

**Comments to the Author**

1. If the authors have adequately addressed your comments raised in a previous round of review and you feel that this manuscript is now acceptable for publication, you may indicate that here to bypass the “Comments to the Author” section, enter your conflict of interest statement in the “Confidential to Editor” section, and submit your "Accept" recommendation.

Reviewer #2: All comments have been addressed

2. Does this manuscript meet PLOS Global Public Health’s publication criteria? Is the manuscript technically sound, and do the data support the conclusions? The manuscript must describe methodologically and ethically rigorous research with conclusions that are appropriately drawn based on the data presented.

Reviewer #2: Yes

3. Has the statistical analysis been performed appropriately and rigorously?

Reviewer #2: Yes

4. Have the authors made all data underlying the findings in their manuscript fully available (please refer to the Data Availability Statement at the start of the manuscript PDF file)?

Reviewer #2: Yes

5. Is the manuscript presented in an intelligible fashion and written in standard English?

Reviewer #2: Yes

6. Review Comments to the Author

Reviewer #2: My comments were addressed in the revision.

7. PLOS authors have the option to publish the peer review history of their article (what does this mean?). If published, this will include your full peer review and any attached files.

**Do you want your identity to be public for this peer review?** For information about this choice, including consent withdrawal, please see our Privacy Policy.

Reviewer #2: **Yes: **Robert Kubinec
